# Stoichiometric Soil Microbial and Enzymatic Characteristics under Three Different Plantation Types in China's Luya Mountain

Xuerong Wang [1], Mengyao Zheng [1], Yue Zhang [1], Ying Chen [2], Lijuan Zhao [3], Baofeng Chai [1] and Tong Jia [1,*]

[1] Shanxi Key Laboratory of Ecological Restoration on Loess Plateau, Institute of Loess Plateau, Shanxi University, Taiyuan 030006, China

[2] School of Environment and Resource Science, Shanxi University, Taiyuan 030006, China

[3] School of Chemistry, Xi'an Jiaotong University, Xi'an 710049, China

[*] Correspondence: jiatong@sxu.edu.cn; Tel.: +86-155-1369-445

**Abstract:** It is important to maintain soil ecosystem function and ecological balance stability. This study uses ecological stoichiometry to ascertain relational constraints of soil nutrient (i.e., carbon (C), nitrogen (N), phosphorus (P), etc.) cycling mechanisms and associated ecological balance characteristics in China's temperate Luya Mountain Nature Reserve. To clarify changes and driving factors associated with soil and extracellular enzyme stoichiometry under different plantation types in July 2021, we analyzed soil nutrient, soil extracellular enzyme, and soil microbial stoichiometry characteristics and their key influencing factors in a *Picea asperata* Mast. forest, a *Caragana jubata* (Pall.) Poir. shrubland, and a *Carex lanceolata* Boott meadow in this reserve. Results revealed significant differences in soil physical and chemical properties, microbial biomass, soil extracellular enzyme activity, and stoichiometry among these different plantation types. Compared to the shrubland and forest plantations, meadow plantation soil was more severely C restricted while that of all three plantations was more N restricted. The main influencing soil stoichiometric ratios were total carbon (TC), total nitrogen (TN), total phosphorus (TP), ammonium ($NH_4^+$-N), soil water content (SWC), β-glucosidase, and microbial C, N, and P biomass. Effects associated with soil N:P, enzymatic N:P, enzymatic C:P, microbial C:N, microbial N:P, and microbial C:P ratios were important for bacterial and fungal community soil structure. This study provides a scientific basis to explicate microbial and regulatory effects of soil extracellular enzyme stoichiometry under different plantation types in one of China's best preserved and most concentrated natural secondary forests.

**Keywords:** stoichiometry; soil extracellular enzyme; microbial biomass; plantation types; Luya Mountain





## 1. Introduction

Ecological stoichiometry is a unified theory that explores the balance between distinct nutrients within ecosystems and how they affect the surrounding ecology. It is primarily concerned with the proportion of carbon (C), nitrogen (N), and phosphorus (P) elements and their shared relationships. This is important for maintaining the elemental balance of ecosystems while also revealing their ecological processes, such as nutrient cycling and restrictions within the ecosystem [1]. Moreover, the relationship between soil and plant C, N, and P ratios can be used to determine nutrient statuses and microbial limitations [2]. It can also be used to characterize nutrient limitations of ecosystems as well as the degree of organic matter decomposition and its potential contribution to soil fertility [3,4]. Soil extracellular enzyme stoichiometry (EES) refers to the extracellular enzyme activity (EEA) ratio related to nutrient acquisition (i.e., C, N, and P), which can be used to reflect microbial resource demands [5]. Moreover, EEAs can be classified as follows: those related to C-acquiring enzymatic activities, N-acquiring enzymatic activities, and P-acquiring enzymatic activities.

Soil microbes are key biological constituents, and their associated EEAs are important driving factors for biogeochemical cycling in terrestrial ecosystems [6], thus playing a critical role in the decomposition, mineralization, and nutrient cycling processes of soil organic matter (SOM) [7]. Soil microbes produce a variety of extracellular enzymes that regulate SOM decomposition, thus driving nutrient cycling processes [8,9]. Additionally, soil microbial biomass carbon (MBC), microbial nitrogen biomass (MBN), and microbial phosphorus biomass (MBP) are important indicators of microbial biomass, which can reveal the total amount of C, N, and P from all living microorganisms in soil. Specifically, MBC and MBN account for 5% of total soil organic carbon (SOC) and N [10,11]. Microbial biomass is highly dynamic and vital for SOM formation and nutrient transport. Moreover, soil microbes are the main driving factors behind C and nutrient release in plant litter and organic matter, which mainly depend on soil EEA activities [12,13]. For example, one study showed that N and P fertilization is conducive to microbial growth in the soil of *Vicia sativa*, where soil microbes responded to soil nutrient changes by regulating the C, N, and P content responsible for in vivo and exocrine extracellular enzyme secretions [14].

Findings from studies on soil extracellular enzymes and associated stoichiometric characteristics closely correlate to soil nutrient cycling processes. Soil extracellular enzymes represent approximate factors of organic matter decomposition, and their activities can be used as indicators of microbial nutrient requirements [2]. Almost all biochemical reactions require enzymes as a catalyst [15]. Soil extracellular enzymes participate in nutrient (i.e., C, N, P, sulfur [S], etc.) cycling processes, and these processes affect associated soil ecological and nutrient transformation cycling processes within the soil system [16,17].

They also represent the key link between plants and soil nutrients. One study showed that plants regulated EESs in subalpine forest soil on the Qinghai–Tibet Plateau, East China [18]. Xiao et al. [19] found that at the regional scale, stoichiometric ratios of soil extracellular enzymes will vary significantly depending on soil texture, environmental condition, and the specific biology. Chen et al. [20] found that total carbon (TC), total nitrogen (TN), and total phosphorus (TP) content as well as microbial and enzymatic C, N, and P content and associated stoichiometric activities in a forest ecosystem changed under soil depth and latitude.

The Luya Mountain National Nature Reserve in Ningwu County, Xinzhou City, Shanxi Province, China, is rich in plant resources, subsequently playing a key role in water conservation, C fixation, oxygen (O) release, nutrient accumulation (i.e., TN and TP), and soil conservation and biodiversity protection measures. These factors are highly significant for maintaining the functional stability and ecological balance of ecosystems [21]. Accordingly, here we tested the soil from three different plantation types (i.e., a *Picea asperata* forest, a *Caragana jubata* shrubland, and a *Carex lanceolata* meadow). Soil nutrient, soil EEA, soil microbial biomass, and microbial community characteristics were examined to explore changes in soil EEA and stoichiometric characteristics of the different planation types and potential driving factors. The aim of this study is to provide a theoretical basis for an in-depth understanding of the regulatory role that microbes play within different ecosystems as well as the mechanisms associated with soil enzyme response. This study also provides a scientific basis for improving cultivation practices within plantation forests and the protection of subalpine vegetation and zonal ecosystems.

## 2. Materials and Methods

### 2.1. Study Site

This study was conducted in Luya Mountain Nature Reserve, Ningwu County, Xinzhou City, Shanxi Province, China. Luya Mountain (38°40′–38° 50′ N, 111°50′–112° 00′ E) is the main peak of Shanxi Province's Guanqin Mountain region, where Heyeping is its highest point (i.e., 2772 MASL). The region is under the influence of a warm temperate continental climate where summers are cool and rainy and winters are cold and dry. According to observational data obtained from the meteorological station in Wuzhai County, Shanxi Province (1401 MASL), local average annual temperatures range from 4 to 7 °C, annual

precipitation ranges from 384 to 679 mm, annual evaporation is 1800 mm, and the frost-free period is between 130 and 170 d. The main vegetation types in the study area include subalpine meadow, boreal temperate coniferous forest, coniferous mixed forest, and shrub meadow [22]. The sampling site information in Luya Mountain is shown in Table 1.

**Table 1.** The sampling sites information in Luya Mountain.

| Vegetation Type | Geographic Coordinates | Altitude/m | Aspect | Slope/($°$) |
|---|---|---|---|---|
| *Picea asperata* forest | 111°54″14″ E 38°44′26″ N | 2280 | Semi-negative slope | 9.8° |
| *Caragana jubata* shrub | 111°50′22″ E 38°43′43″ N | 2750 | South slope | 0° |
| *Carex lanceolata* meadow | 111°50′22″ E 38°43′43″ N | 2750 | South slope | 0° |

### 2.2. Soil Sample Collection and Measurements

In July 2021, an S-type sampling method was adopted for this study [23]. Five sampling points were selected in each plantation type after leaf and plant debris were removed. Using a soil drill, samples were collected from the 0–20 cm soil layer. Soil pH (1:25 soil-water ratio) was measured using the point-based soil sampling method. An elemental analyzer (vario EL/MACRO cube, Elementar, Hanau, Germany) was used to determine TC and TN content in soil while the molybdenum blue colorimetric method was used to determine TP. Refer to the information on soil agrochemical analysis for more details on the methods used in this study. A total organic carbon element analyzer (vario TOC, Elementar, Hanau, Germany) was used to determine SOC [24].

### 2.3. Soil Enzyme Activities and Microbial Nutrient Elements

This study used the enzyme-linked immunoassay assay (ELISA) to detect four enzyme types associated with soil C, N, and P cycles: N-acetyl-β-D-glucosidase (S-NAG), β-glucosidase (β-GC), leucine aminopeptidase (L-LAP), and soil neutral phosphatase (NP). Two additional enzymes (i.e., polyphenol oxidase (PPO) and peroxidase (POD)) were also used. All enzyme types were determined at Shenggong Biological CO, LTD. After completing chloroform fumigation extraction, MBC, MBN, and MBP were extracted by potassium sulfate (TOC-L), potassium sulfate (flow analyzer), and sodium bicarbonate (molybdomancy and colorimetric resistance), respectively [25,26].

### 2.4. Microbial Sequencing Technology

Soil microbial community characteristics were obtained through extraction, polymerase chain reaction (PCR) amplification, and microbial DNA sequencing. Universal primers were used to amplify bacterial 16S rRNA and fungal internal transcribed spacer (ITS) genes. Specifically, we used ITS1F (5′-CTTGGTCATTTAGAGGAAGTAA) and ITS2R (5′-GCTGCGTTCTTCATCGAT-3′) as fungal primers and 338F (5′-AACMGGATTAGATACCCKG) and 806R (5′-ACGTCATCCCCACCTTCC-3′) as bacterial primers. Finally, sequencing was performed on the MiSep platform (Illumina, Inc., San Diego, CA, USA) at Shanghai Magi Biomedical Technology Co., LTD [27]. We submitted the raw sequencing data to the National Center for Biotechnology Information (NCBI) Sequence Read Archive (SRA) (https://www.ncbi.nlm.nih.gov/sra, accessed on 7 February 2023) under project accession number PRJNA932581.

### 2.5. Data Analyses

QIIME software was used to integrate the original sequencing data in FASTQ format. All chimeras in the sequences were removed. UPARSE operational taxonomic unit (OTU) clustering software (version 11, http://www.drive5.com/uparse/, accessed on 7 February 2023) was used to attain high-quality sequence clustering operations according to 97% taxa

similarity. The OTU sequences were then classified and compared using the Bayesian RDP classifier algorithm (version 2.11, SourceForge Headquarters, San Diego, CA, USA). The SILVA v138 database was used for the bacterial sequence database, while the UNITE v8.0 database was used for the fungal sequence database. The reliability threshold was 70%. Finally, according to the minimum sample sequence number, the OTU table of the reserved and conserved microbial communities was generated.

Vector analysis (i.e., vector length (VL) and vector angle (VA)) was used to describe soil nutrient element limitations. VL is associated with soil C limitation (where the greater the VL is, the greater soil C will be). VA is associated with N/P limitation. In other words, when VA is greater than 45°, soil P limitation will be greater than soil N limitation; when VA is less than 45°, soil N limitation will be greater than soil P limitation. The following formulae were used to calculate VL and VA[28]:

$$\text{Vector Length (L)} = \{[\ln BG/\ln (NAG + LAP)]2 + (\ln BG/\ln NP)2\}1/2 \tag{1}$$

$$\text{Vector angle (A)} = \text{Degrees} \{ATAN2 [(\ln BG/\ln NP), (\ln BG/\ln (NAG + LAP))]\} \tag{2}$$

where S-NAG denotes N-acetyl-β-D-glucosidase (IU/L), β-GC denotes β-glucosidase (IU/L), L-lap denotes leucine aminopeptidase (IU/L), and NP denotes neutral phosphatase (IU/L).

### 2.6. Statistical Analysis

One-way ANOVA was used to examine plant type effects on soil biochemical and ecological enzyme parameters. Duncan's multi-range test ($p < 0.05$) was used for comparison. SigmaPlot v14.0 (Systat Software Inc., San Jose, CA, USA) was used to visualize data analysis results. Pearson correlation coefficients were used to analyze correlations between soil physicochemical properties and stoichiometric ratios. SPSS version 20.0 (IBM, Chicago, IL, USA) was used for one-way ANOVA, while Canoco v4.5 (Microcomputer Power, Ithaca, NY, USA) was used for redundancy analysis (RDA).

## 3. Results

### 3.1. Soil Physiochemical Properties

Soil physiochemical properties of the *P. asperata* forest, the *C. jubata* shrubland, and the *C. lanceolata* meadow plantations significantly differed. Among these properties, we observed a significant difference between the forest and meadow plantations, where forest soil moisture content was higher by a factor of 2.9% compared to that of the meadow plantation (Table 2). Additionally, significant soil TN differences were observed among the three plantation types ($p < 0.05$), which were as follows: shrub > meadow > forest (5.801, 4.809, and 3.406, respectively). Significant differences were also observed in soil TP ($p < 0.05$): meadow > shrub > forest. Moreover, the $NH_4^+$-N content of the three plantation types significantly differed ($p < 0.05$), where $NH_4^+$-N content in the meadow and forest plantations was 27.01% and 32.29% higher compared to the shrubland plantation, respectively (Table 2).

### 3.2. Variation in Soil Enzyme Activities

Significant differences were observed in N-acetyl-β-D-glucosidase (S-NAG), β-glucosidase (β-GC), polyphenol oxidase (PPO), peroxidase (POD), leucine aminopeptidase (L-LAP), and neutral phosphatase (NP) enzyme actives in the three plantation types (Figure 1A). Shrubland N-acetyl-β-D-glucosidase (S-NAG) enzyme activity was highest (119.536 IU/L), namely, greater by factors of 1.07 and 1.06 compared to the meadow and forest plantations, respectively. Moreover, β-glucosidase (β-GC) activity significantly differed among these different soil types, where it was highest in the meadow plantation (39.392 IU/L), being 4.152 IU/L higher than the forest plantation. PPO activity significantly differed among the three plantation types, where it was highest in the shrubland plantation (249.508 IU/L) and lowest in the forest plantation (28.899 IU/L). POD activity also significantly differed among

the three plantation types, where it was highest in the shrubland plantation (15.451 mU/L) and lowest in the meadow plantation (13.141 mU/L), namely, by a factor of 17.58%. Leucine aminopeptidase (L-LAP) activity was significantly higher in the meadow and forest plantations than in the shrubland plantation. Neutral phosphatase (NP) had the opposite effect to leucine aminopeptidase (L-LAP), where it was significantly higher in the shrubland plantation than in the forest plantation (i.e., 42.646 IU/L) (Figure 1A).

**Table 2.** Physical and chemical properties of soil under three planting cover types.

|  | C | G | L |
| --- | --- | --- | --- |
| SWC (%) | 0.028 ± 0.003 b | 0.039 ± 0.002 ab | 0.057 ± 0.026 a |
| pH | 6.528 ± 0.276 a | 6.715 ± 0.265 a | 6.505 ± 0.177 a |
| TN (g/kg) | 4.809 ± 0.566 b | 5.801 ± 0.782 a | 3.406 ± 0.617 c |
| TC (g/kg) | 53.493 ± 5.56 a | 64.356 ± 9.249 a | 57.573 ± 13.487 a |
| SOC (g/kg) | 51.689 ± 5.46 a | 62.238 ± 9.25 a | 55.524 ± 12.96 a |
| TP (g/kg) | 0.956 ± 0.043 a | 0.826 ± 0.118 b | 0.638 ± 0.067 c |
| SOM (g/kg) | 89.111 ± 9.414 a | 107.299 ± 15.947 a | 95.723 ± 22.343 a |
| $NO_3^-$-N (mg/kg) | 1.234 ± 0.426 a | 1.245 ± 0.462 a | 1.136 ± 0.084 a |
| $NH_4^+$-N (mg/kg) | 31.645 ± 4.511 a | 24.915 ± 3.326 b | 32.96 ± 4.3 a |

Note: C, G and L represent different vegetation types, C represents *Carex lanceolat* meadow, G represents *Caragana jubata* shrubby, L represents *Picea Asperata* forest, and same below. SWC represents soil moisture content, pH represents soil pH, TN represents soil total nitrogen, TC represents soil total carbon, SOC represents soil organic carbon, TP represents soil total phosphorus, SOM represents soil organic matter, $NO_3^-$-N represents soil nitrate nitrogen, and $NH_4^+$-N represents soil ammonium nitrogen. Different lowercase letters indicate significant differences according to Duncan's test ($p < 0.05$).

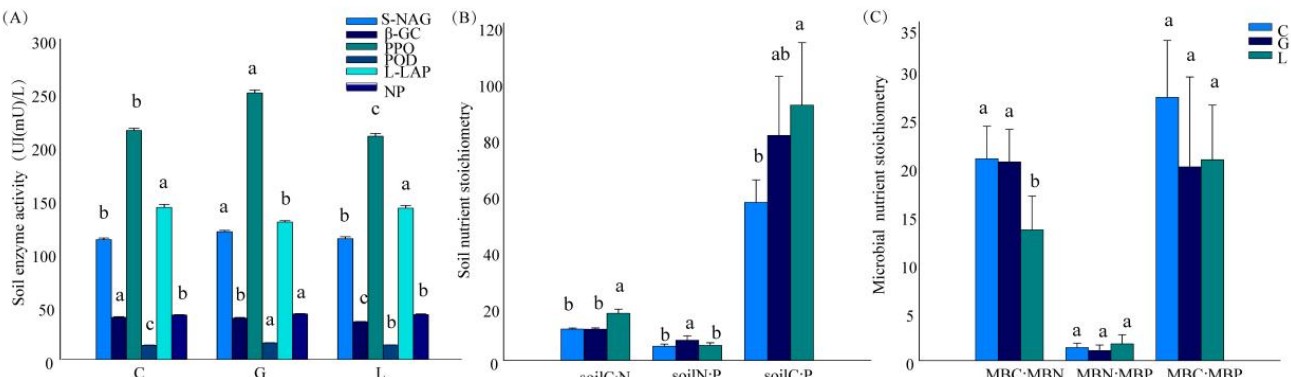

**Figure 1.** Stoichiometric characteristics of soil enzyme activities (**A**), soil nutrients (**B**) and microbial biomass (**C**) under three planting cover types. Different lowercase letters indicate significant differences according to Duncan's test ($p < 0.05$).

### 3.3. Stoichiometric Characteristics of Soil and Microbial Nutrients

The stoichiometry of soil and microbial nutrients differed among the different plantation types. The MBC:MBN ratio between the meadow and shrubland plantations was higher by a factor of 1.5 compared to the forest plantation, while no significant differences were observed between MBN:MBP and MBC:MBP ratios among the three plantation types. Meadow and forest soil N:P ratios were significantly lower compared to that of the shrubland plantation (5.05, 5.35, and 7.17, respectively). Soil C:P ratios among the meadow, shrubland, and forest plantations significantly differed, namely, being highest in the forest and lowest in the meadow plantations (i.e., 61.53% higher than the meadow plantation). Additionally, the forest soil C:N ratio was 50.35% higher than meadow and shrubland plantation values, and the difference was significant ($p < 0.05$). The forest soil C:P ratio was significantly higher than that of the meadow, reaching a maximum of 108.737:1 (Figure 1B,C).

### 3.4. Correlation Analysis of Soil Extracellular Enzymes and Microbial Biomass Stoichiometry with Environmental Factors

The MBC:MBN ratio positively correlated to the soil enzyme stoichiometric C:N ratio ($p < 0.05$) and significantly negatively correlated to the soil C:N ($p < 0.01$) and C:P ($p < 0.05$) ratios. The soil C:N ratio negatively correlated to the enzyme C:N and C:P ratios, and the soil N:P ratio significantly negatively correlated to the enzyme N:P ratio. In other words, we observed a significant negative correlation between stoichiometric soil nutrient ratios and corresponding enzyme activity elemental ratios ($p < 0.05$) (Figure 2). We conducted RDA on stoichiometric ratios by classifying and selecting environmental factors before and after selecting the remaining environmental factors. Results showed that TC, TN, TP, $NH_4^+$-N, soil water content (SWC), β-GC, MBC, MBN, and MBP were the main influencing factors for the stoichiometric ratios. RDA showed that the explanatory degree of soil physicochemical factors on stoichiometric ratios was 85.7% on the first axis, 2.8% on the second axis, and 88.5% for the two axes combined (Figure 3A). Additionally, RDA showed that the explanatory degree of microbial biomass nutrients and soil EEAs on stoichiometric ratios was 72.4% on the first axis, 9.3% on the second axis, and 91.7% for the two axes combined (Figure 3B). Soil TN positively correlated to TC and TP, while TC and TP were negatively correlated (Figure 4).

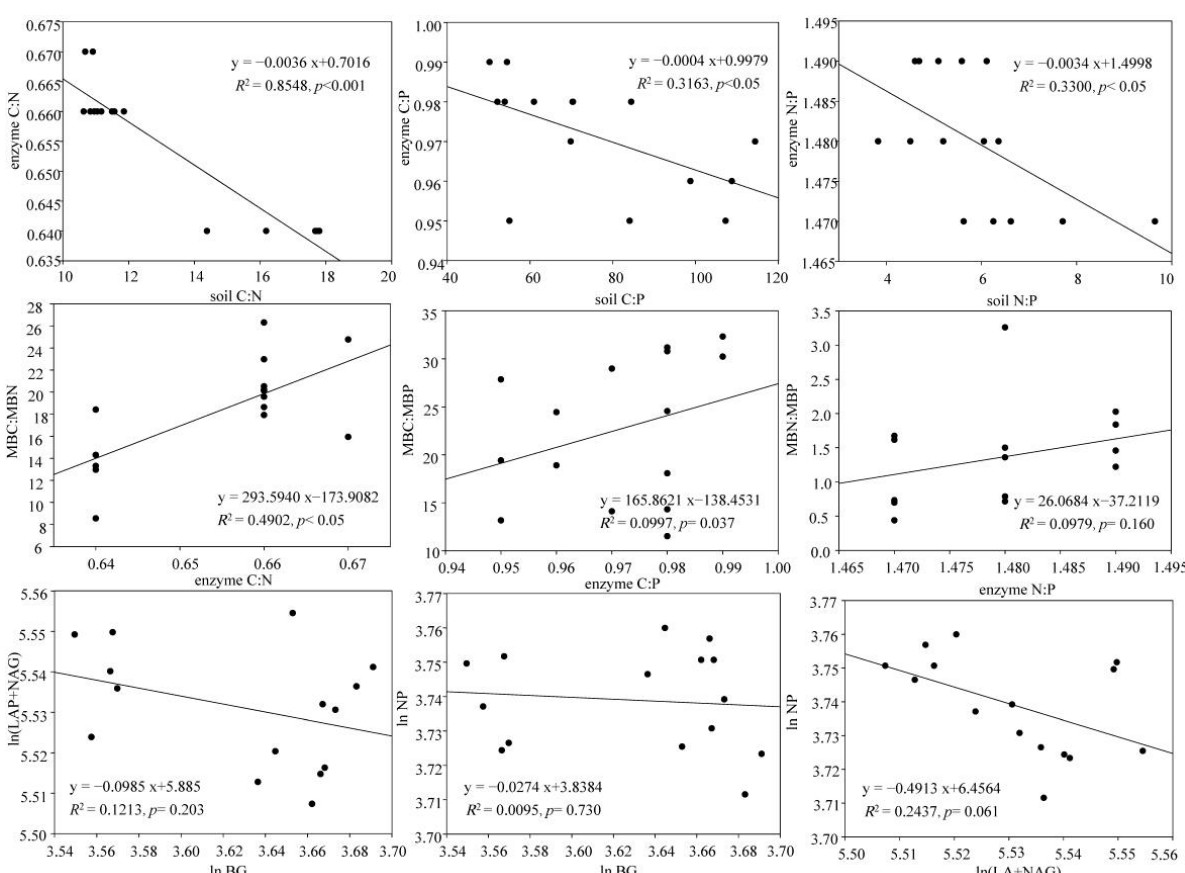

**Figure 2.** Correlation analysis of soil extracellular enzyme stoichiometry with soil C, N and P or microbial biomass C, N and P stoichiometry.

### 3.5. Ecoenzymatic Stoichiometry

Results showed that VL and VA significantly differed under the three different plantation types ($p < 0.05$) (Table 3). For example, the VL of soil extracellular enzymes decreased incrementally among the meadow (*C. lanceolata*), the shrubland (*C. jubata*), and the forest (*P. asperata*) plantations, indicating that C soil restrictions were highest in meadow soil, followed by shrubland and forest soil. Under these the plantation types, VA was less throughout (by 45°), indicating that these three soil types were mainly limited by N. Among

them, the relative the degree of N restriction of the shrubland plantation was smallest and significantly lower than meadow and forest soil ($p < 0.05$).

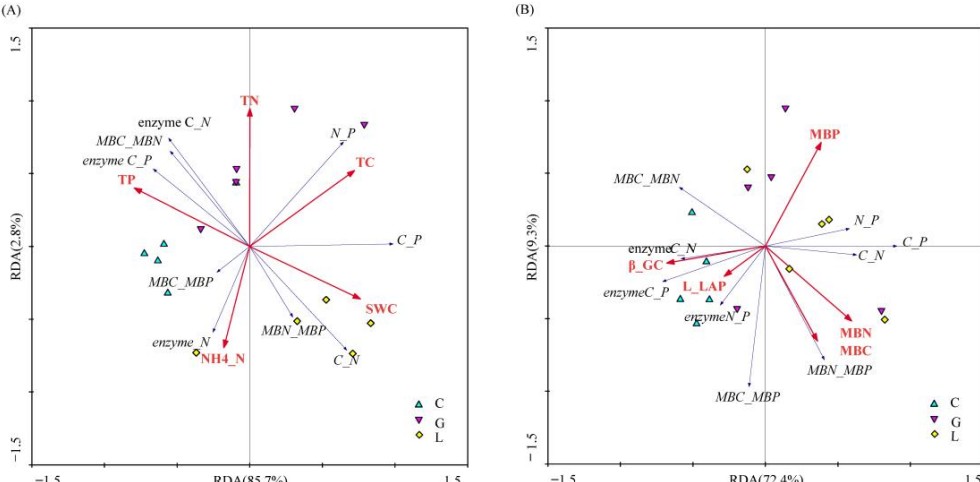

**Figure 3.** Redundancy analysis of the stoichiometric ratios of soil nutrients (**A**), microbial nutrients and soil extracellular enzyme activities (**B**) of three planting coverts.

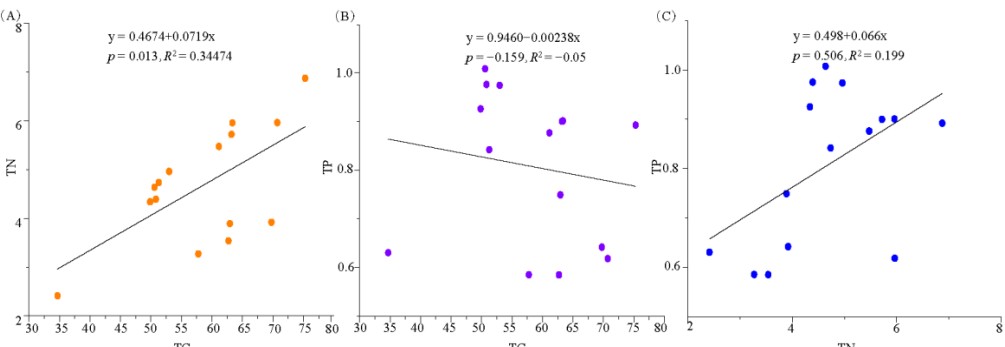

**Figure 4.** Correlation between soil total carbon and nitrogen (**A**), soil total carbon and phosphorus (**B**), and soil total nitrogen and phosphorus (**C**).

**Table 3.** Stoichiometric vector analysis of three planting cover types of soil. Different lowercase letters indicate significant differences according to Duncan's test ($p < 0.05$).

|  | Vector L | Vector A |
|---|---|---|
| C | 1.19 ± 0.006 a | 33.93 ± 0.097 b |
| G | 1.18 ± 0.005 b | 34.24 ± 0.030 a |
| L | 1.15 ± 0.005 c | 34.01 ± 0.075 b |

### 3.6. Characteristics of Soil Bacterial and Fungal Communities under the Different Plantation Types

#### 3.6.1. Soil Bacterial and Fungal Community Diversity of the Different Plantation Types

After quality screening and minimum sample sequence number extraction, the resulting bacterial or fungal effective sequences were used for analysis, which were clustered into 4847 bacterial OTUs and 2630 fungal OTUs according to 97% similarity.

Results from this study showed that there were significant differences in soil bacterial and fungal community alpha (α) diversity in the *C. lanceolata* meadow, the *C. jubata* shrubland, and the *P. asperata* forest plantations ($p < 0.05$): The Shannon index revealed significant soil bacterial community differences between the meadow and shrubland plantations, while the Simpson index revealed significant soil bacterial community differences between the shrubland and forest plantations, among which Simpson index soil bacterial

community results were the highest. The Sobs, Ace, Chao1, and Coverage indexes between the shrubland and meadow plantations (with the lowest overall values) significantly differed from that of the forest soil fungal community (Table 4, $p < 0.05$).

**Table 4.** Variation characteristics of α diversity of bacterial and fungal communities in three planted cover soils. Different lowercase letters indicate significant differences according to Duncan's test ($p < 0.05$).

|  |  | C | G | L |
|---|---|---|---|---|
| Sobs | bacteria | 1808.000 ± 135.220 a | 1903.200 ± 98.210 a | 1801.200 ± 89.256 a |
|  | fungi | 299.800 ± 89.433 b | 424.200 ± 122.608 ab | 524.600 ± 59.622 a |
| Shannon | bacteria | 6.000 ± 0.093 b | 6.238 ± 0.104 a | 6.120 ± 0.108 ab |
|  | fungi | 3.638 ± 1.094 a | 3.704 ± 0.969 a | 3.589 ± 0.390 a |
| Simpson | bacteria | 0.007 ± 0.001 a | 0.005 ± 0.001 c | 0.006 ± 0.001 b |
|  | fungi | 0.131 ± 0.196 a | 0.124 ± 0.155 a | 0.072 ± 0.019 a |
| Ace | bacteria | 2843.900 ± 504.220 a | 2691.909 ± 104.928 a | 2622.585 ± 173.886 a |
|  | fungi | 311.215 ± 91.264 b | 451.525 ± 144.021 b | 595.417 ± 58.499 a |
| Chao | bacteria | 2623.274 ± 267.273 a | 2690.841 ± 71.938 a | 2539.597 ± 115.243 a |
|  | fungi | 313.440 ± 92.766 b | 457.557 ± 149.277 ab | 593.096 ± 57.298 a |
| Coverage | bacteria | 0.966 ± 0.004 a | 0.966 ± 0.001 a | 0.968 ± 0.001 a |
|  | fungi | 1.000 ± 0 a | 0.999 ± 0.001 a | 0.998 ± 0.001 b |

### 3.6.2. Soil Bacterial and Fungal Community Structure under the Different Plantation Types

Based on Bray–Curtis dissimilarity, non-metric multidimensional scaling (NMDS) was applied to the fungal and bacterial communities of the different plantation types at an OUT scale. Results showed that plantation types had a significant effect on fungal and bacterial community structure ($p < 0.001$). Moreover, the different plantation types exhibited both uniform and significant effects on soil bacterial and fungal communities (bacteria: stress = 0.073, $R^2 = 0.408$, $p = 0.001$; fungus: stress = 0.094, $R^2 = 0.295$, $p = 0.001$; Figure 5).

### 3.6.3. Soil Microbial Community Composition

Under all three plantation types (i.e., meadow, shrubland, and forest), Actinomycetota, Pseudomonadota, Acidobacteriota, and Chloroflexota comprised the dominant bacterial community (Figure 6A). These four phyla accounted for 74% of the soil bacterial community of the *C. lanceolata* meadow, 78% of the soil bacterial community of the *C. jubata* shrubland, and 83% of the soil bacterial community of the *P. asperata* forest. The dominant fungal community species members (Figure 6B) were from the Ascomycota and Basidiomycota phyla, accounting for 75%, 78%, and 85%, respectively.

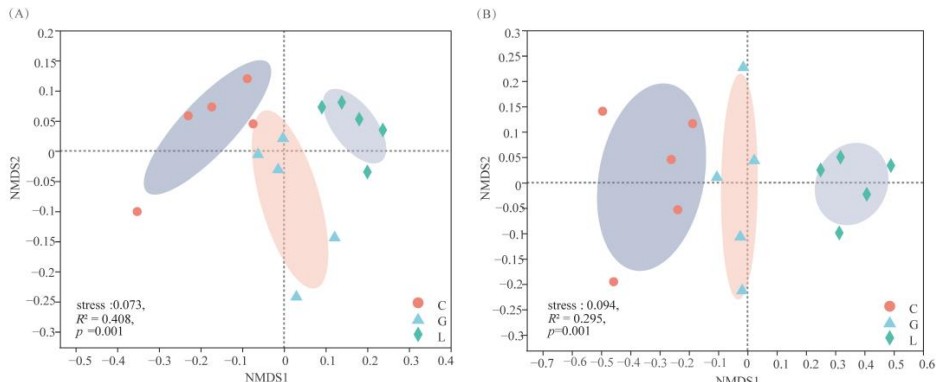

**Figure 5.** Community structure characteristics of soil bacteria (**A**) and fungi (**B**) under three planting coverts.

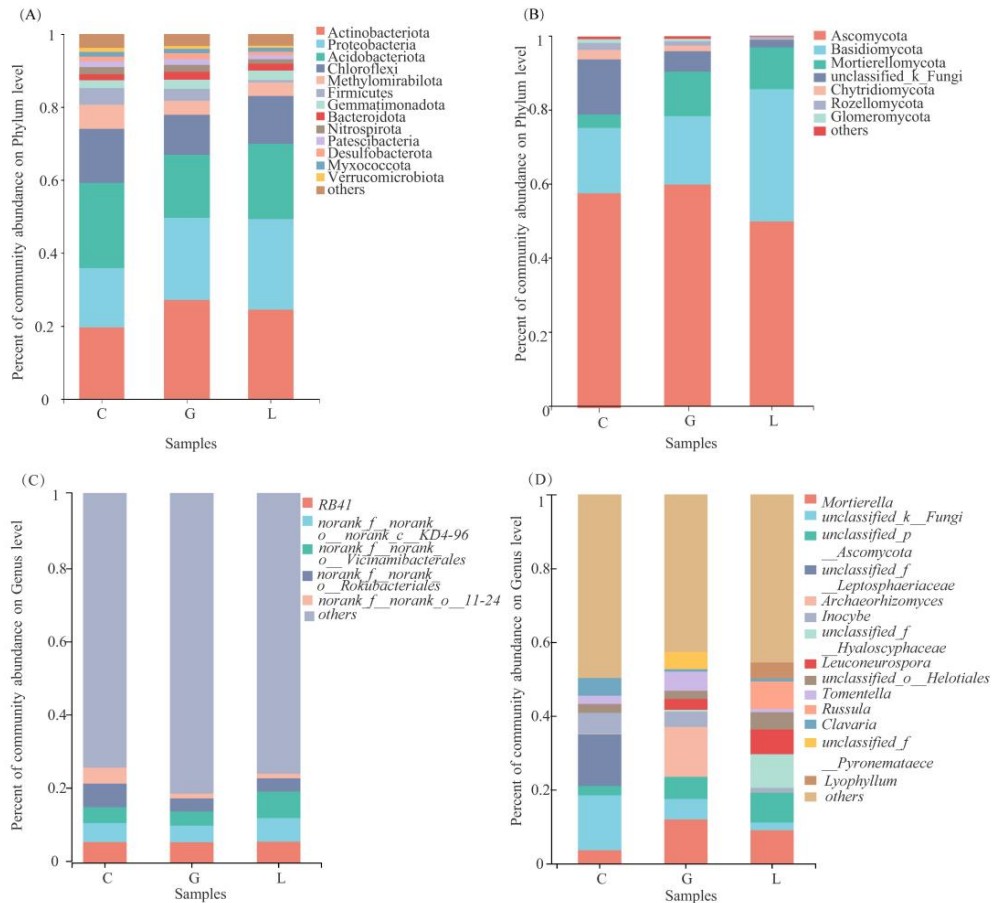

**Figure 6.** (**A**,**B**) Bacteria (**A**) and fungi (**B**) with relative abundance greater than 1% at phylum level in three planting coverts; (**C**,**D**) bacteria (**C**) and fungi with relative abundances greater than 4% at genus level in the three planted coverts (**D**).

### 3.6.4. Analysis of Bacterial and Fungal Community Composition in the Different Plantation Types

At the genus level, the dominant soil bacterial and fungal community groups of the three plantation types significantly differed ($p < 0.05$) (Figure 7). *Bacillus* abundance in the soil bacterial community of the *P. asperata* forest plantation significantly differed from the relative *Bacillus* abundance in the soil of the *C. lanceolata* meadow ($p < 0.001$) and the *C. jubata* shrubland ($p < 0.05$) plantations (Figure 7A). Contrary to the relative distribution of *Bacillus* abundance, the relative *Bradyrhizobium* abundance was highest in *P. asperata* forest soil, which significantly differed from *C. lanceolata* meadow soil ($p < 0.01$) and *C. jubata* shrubland soil ($p < 0.05$) (Figure 7A). Moreover, the relative abundance of *Lipomyces* in forest soil was significantly higher than that of meadow and shrubland soil ($p < 0.05$). The relative abundance of *Archaeorhizomyces* was highest in shrubland soil ($p < 0.05$) (Figure 7B).

### 3.7. Correlation between Soil Enzyme Stoichiometry and Dominant Species

Actinomycetota and Pseudomonadota significantly and positively correlated with the soil C:P ratio ($p < 0.01$), while Actinomycetota significantly and negatively correlated with the enzyme N:P ratio ($p < 0.01$). There was a significant negative correlation between Pseudomonadota and the enzyme N:P ratio ($p < 0.05$) and a significant positive correlation between Firmicutes and soil enzyme C:P and N:P ratios ($p < 0.001$). (Figure 7C). Similarly, soil fungal communities under the different plantation types were also significantly affected by stoichiometric ratios (Figure 7D). Basidiomycota positively correlated with the soil C:N ratio ($p < 0.01$) and inversely correlated with the enzyme C:N ratio ($p < 0.01$), the enzyme C:P ratio ($p < 0.01$), and the microbial C:N ratio ($p < 0.05$). In contrast, Chytridiomycota

positively correlated with the enzyme C:N ratio ($p < 0.01$), the enzyme C:P ratio ($p < 0.01$), and the microbial C:N ratio, while negatively correlating with the soil C:N ratio (Figure 7D).

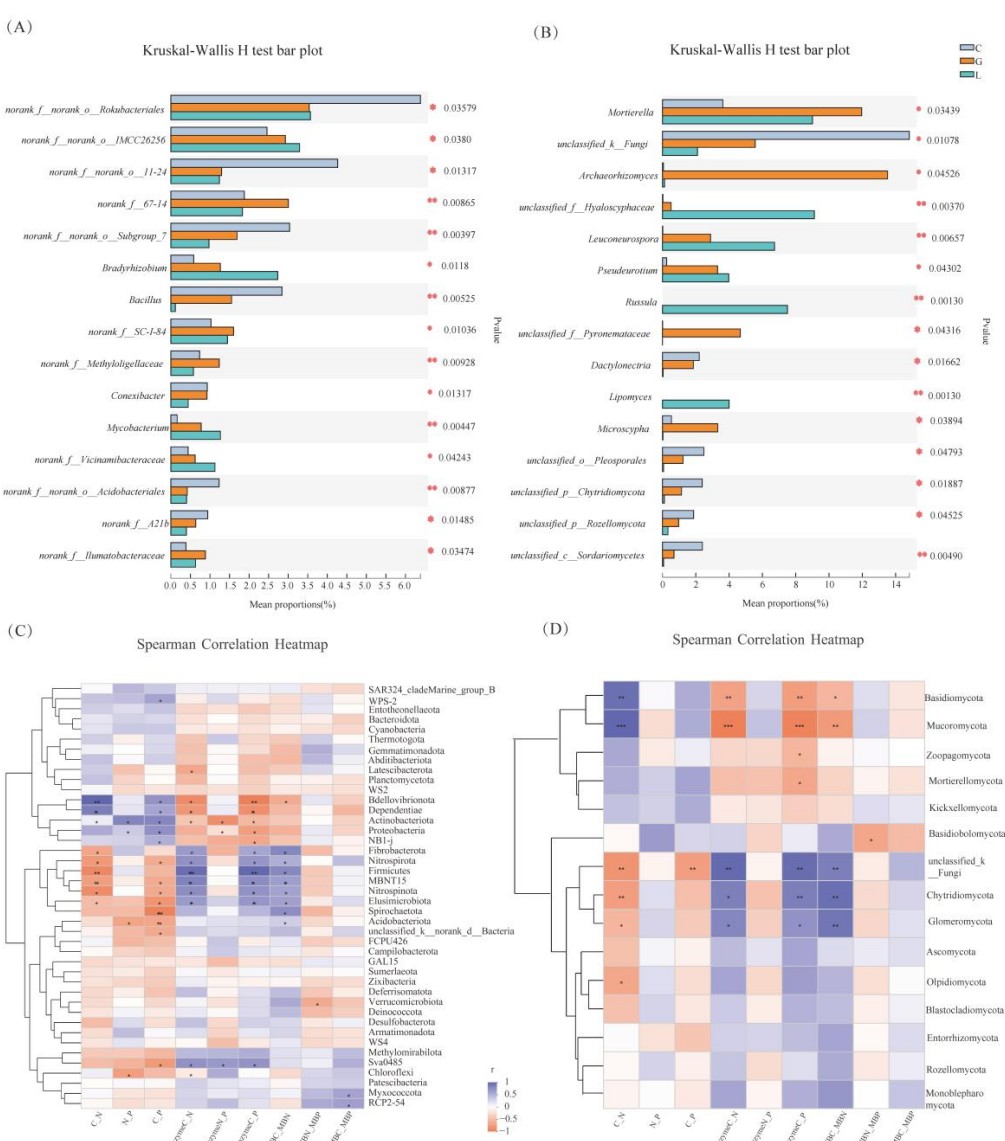

**Figure 7.** The difference in dominant bacteria (**A**) and dominant fungi (**B**) (top 15 relative abundances) in three planting coverts and the correlation analysis of dominant bacteria (**C**) and dominant fungi (**D**) with the stoichiometric ratio of soil nutrients, extracellular enzymes and microbial nutrients. $0.01 < p \leq 0.05$ *, $0.001 < p < 0.01$ **, $p < 0.001$ ***.

### 3.8. Stoichiometry Relative to Environmental Factors and Dominant Microbial Groups

We analyzed correlations between soil, soil microbes, and soil enzyme stoichiometry and soil bacterial and fungal communities based on RDA analysis (Figure 8). Results showed that the first two axes of the soil bacterial community accounted for 46.62% (Figure 8A), while results from canonical correspondence analysis (CCA) showed that the first two axes of the soil fungal community accounted for 20.49% of total variation (Figure 8B). Variance inflation factor (VIF) screening revealed that stoichiometric ratios (i.e., soil N:P, enzymatic N:P, enzymatic C:P, microbial C:N, microbial N:P, and microbial C:P) had important impacts on soil bacterial and fungal community structure (Figure 8C,D). Among them, the enzyme C:P ratio had the greatest impact on the bacterial community, while the enzyme N:P ratio had the greatest impact on the fungal community. Pertaining to the fungal community, *Basidiomycota*, *Pleosporales*, and *Laccaria* were mainly affected by soil extracellular enzyme C:N ratios. *Dothideomycetes* was most affected by the MBN:MBP ratio.

Both RDA and CCA analysis showed that forest soil microbes were mainly affected by soil C:N ratios, while shrubland soil microbes were more affected by soil N:P ratios.

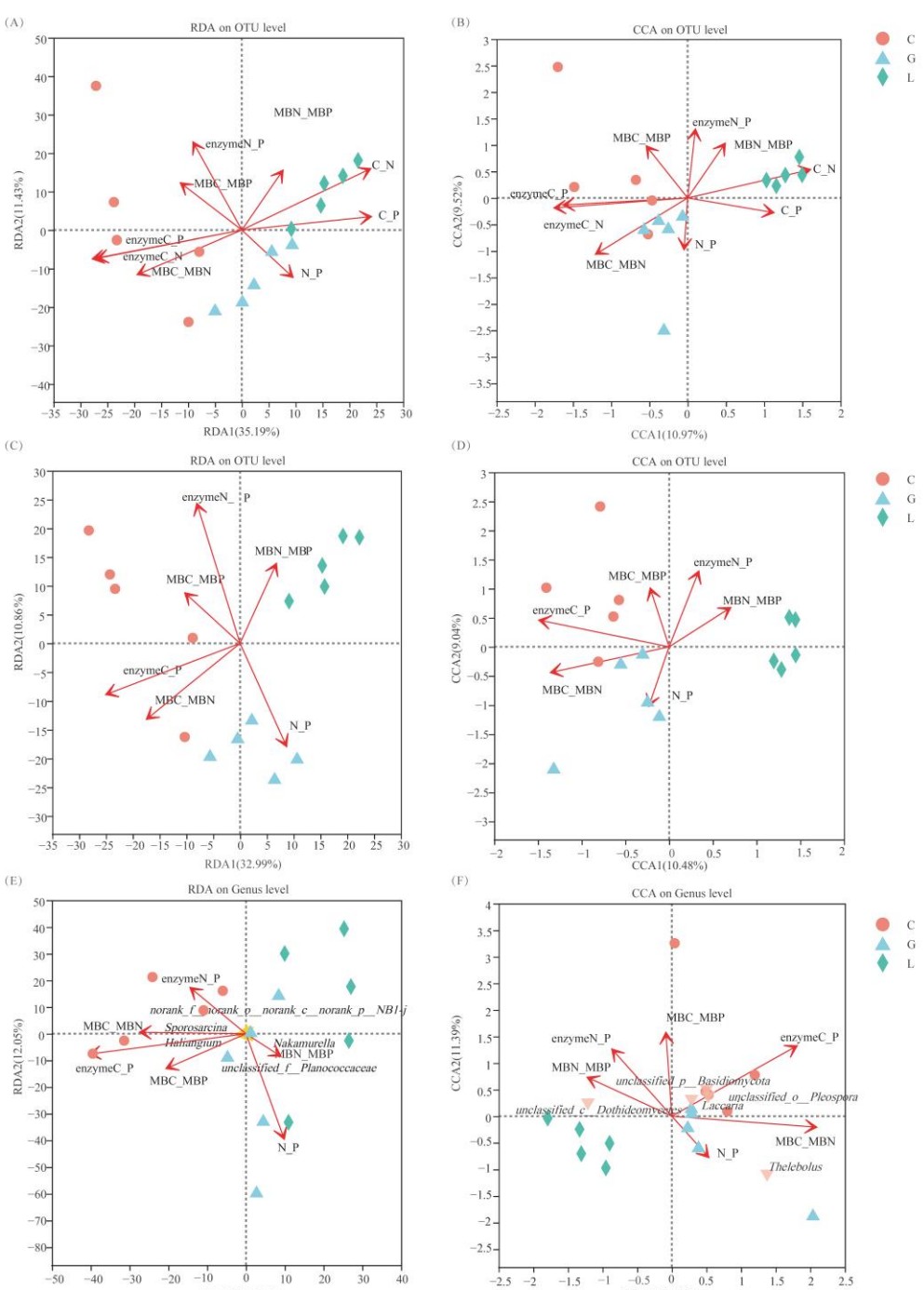

**Figure 8.** (**A**,**B**) The stoichiometric effects of soil, soil microorganisms and soil enzymes on bacterial (**A**) and fungal (**B**) communities and their interactions. (**C**,**D**) The stoichiometric effects and interactions of soil, soil microorganisms and soil enzymes on bacterial (**C**) and fungal (**D**) communities after VIF variance expansion factor screening. (**E**,**F**) Effects and interactions of stoichiometry of soil, soil microorganisms and soil enzymes on dominant genus (top5) of bacteria (**E**) and fungi (**F**) after VIF variance expansion factor screening for three planting subtypes.

## 4. Discussion

### 4.1. Effects of the Different Plantation Types on Soil Nutrients, Enzyme Activities, and Microbial Nutrients

In this study, soil TN and TP significantly differed ($p < 0.05$), while TC and TN significantly correlated under the three plantation types. This is consistent with the results from Wang et al. [29], who observed certain differences in the spatial distribution of soil nutrients among plantation types. The classification and selection of soil extracellular enzymes in our study were identical to that used by Wang et al. [30]. Different vegetation types affect soil enzyme activity differently [31]. In this study, three soil extracellular enzyme types were identified (i.e., 1,4-β-glucosidase as a C harvester, N-acetyl-β-D-glucosidase as an N harvester, and leucine aminopeptidase as an N harvester). Moreover, phosphatase activity significantly differed ($p < 0.05$). Results also showed that soil nutrients correlated with soil extracellular enzyme activity. Studies have previously shown that an increase or a decrease in soil nutrient content can cause changes in soil EEAs [32]. Peng et al. [33] reported that soil enzyme activity and associated stoichiometric effects were impacted by vegetation, climate, and soil factors.

Through our soil nutrient ratio analysis, we found that the soil N:P ratio of the shrubland plantation type was significantly higher than the meadow and forest plantation types ($p < 0.05$, Figure 1); however, the overall N:P ratio was less than 14, which indicated that N restrictions were relatively higher in the study area [34]. At the same time, Moorhead et al. [28] reported that VL correlated to soil C limitations while VA correlated to N and P limitations. In this study, soil VL and VA showed that the soil C limitation to that of N and P in the *C. lanceolata* meadow plantation was greater than that observed in the soil of the *C. jubata* shrubland and the *P. asperata* forest plantations, while soil VA in the (spruce) forest and the (steppe) meadow plantations was significantly higher than that in the shrubland plantation. However, both were less than 45°, indicating that N limitation was greater than P limitation ($p < 0.05$, Table 3).

### 4.2. Relationships among Soil Extracellular Enzymes, Microbial Nutrients, and Soil Nutrient Stoichiometry under the Different Plantation Types

Plants and the soil they grow in are intrinsically related, together forming a whole. Interactions occur among plants, soil, and microorganisms. Plants affect soil and microbes while soil and microbes have a certain effect on the aboveground components of plants, including the relationships between soil and microbes. Soil extracellular enzymes are produced by soil microbes, root exudates, and the decomposition of plant and animal residue [35]. In their study on forest ecosystems in China, Xu et al. [32] observed a negative correlation between soil extracellular enzyme C:N and N:P ratios and soil C:N and N:P ratios. The stoichiometric characteristics of soil were significantly affected by plants and their vegetation types [36]. Our study observed similar correlations. Specifically, except for soil extracellular enzyme C:N and N:P ratios, we observed a significant negative correlation between soil extracellular enzyme N:P ratios and soil N:P ratios ($p < 0.05$). We also found a significant positive correlation between soil extracellular enzyme C:N and microbial C:N ratios ($p < 0.01$, Figure 2).

Through redundancy analysis, we observed that microbial communities significantly correlated to econometric and soil stoichiometric ratios ($p < 0.05$, Figure 8). This may be due to how different plant types influence soil nutrients (as well as other reasons), resulting in different physical and chemical soil properties (Table 2), which has a certain impact on microbial nutrients. For example, in their investigation on changing C, N, and P content and associated stoichiometric effects among soil, microbes, and enzymes in a forest ecosystem (respective to depth and latitude), Chen et al. [20] reported that soil C and other nutrient content may be the determining factor of microbial C and nutrient biomass. Lauber et al. [37] reported that soil bacterial and fungal communities can be affected by soil TP content. This may be related to the different dominant microbial species that reside in the soil of different plant types (Figure 5), resulting in differential microbial exocrine

enzyme secretions and other such chemical substances. This is consistent with results from Wang et al. [38] who reported that plant types and soil properties have significant effects on soil microbial community structure and composition. The composition and relative abundance of microbial communities are likely to be heavily influenced by ecological and soil properties, such as vegetation type, soil texture, soil type, moisture, and nutrient concentrations [39]. Xiao et al. [40] observed differences in soil microbial structure and diversity characteristics in plants growing at different altitudinal zones. Through biologic and abiotic factors, soil bacteria and fungus also have a determining effect on the nutrients they require, thus regulating soil EEA and associated measurements [41]. Additionally, enzyme production will decrease under unlimited microbial resources. However, both energy and extracellular enzymes are preferentially allocated to compensate for losses in the most restricted resources under conditions of limited resources [42].

*4.3. Key Soil Microbial and Soil Extracellular Enzyme Stoichiometry Factors under Different Plantation Types*

There are some studies on soil microbes and soil EES. For example, one such study found that plant diversity and species richness altered soil EES in an arid steppe ecosystem on the Chinese Loess Plateau [43]. German et al. [44] reported that soil temperature may directly affect extracellular enzymes or EEAs through their effect on microbes on the global scale. Moreover, Wang et al. [35] found that stoichiometric effects associated with soil C, N, and P enzyme absorption closely correlated to microbial community composition. Wang et al. [10] revealed that different plant types can significantly affect microbial resource acquisition (i.e., EEA and stoichiometric ratios), while the SOC supply in the substrate was shown to be the key influencing factor. Results from this study showed significant changes in soil enzyme stoichiometry among the different plantation types, while soil microbial community characteristics significantly correlated to soil extracellular enzyme C:P, N:P, and soil C:N ratios. Relevant studies have also shown that soil microbes can regulate soil and associated EES by assimilating soil nutrients [3]. Lin et al. [41] reported that soluble DOC and MBP concentrations in soil are key regulatory factors of soil enzyme activities and associated stoichiometry. In this study, we found that soil TN and TP in the different plantation types significantly differed ($p < 0.05$). Additionally, vector analysis results showed that P was limited in the soil of all three plantation types. RDA of microbial nutrients on stoichiometric characteristics showed that MBP had a significant effect on ecological stoichiometry (Figure 3). Therefore, this study showed that soil TP and microbial P content are extremely important for soil microorganisms and EES.

## 5. Conclusions

Our analysis of soil, soil microbes, and extracellular enzymes and their associated stoichiometric ratios in China's Luya Mountain Nature Reserve showed that the degree of influence among three different plantation types (i.e., a *Picea asperata* forest, a *Caragana jubata* shrubland, and a *Carex lanceolata* meadow) differed in soil properties, soil microbes, and extracellular enzymes and associated stoichiometric ratios, while soil properties, soil microbes, and extracellular enzymes and associated stoichiometric ratios influenced each other. *P. asperata* forest soil moisture content was higher by a factor of 2.9% compared to that of the meadow plantation. $NH_4^+$-N content in the meadow and forest plantations was 27.01% and 32.29% higher compared to the shrubland plantation, respectively. Shrubland N-acetyl-β-D-glucosidase enzyme activity was highest, namely, greater by factors of 1.07 and 1.06 compared to the meadow and forest plantations, respectively. PPO activity significantly differed among the three plantation types, where it was highest in the shrubland plantation and lowest in the forest plantation. The soil fungal communities significantly correlated to the soil extracellular enzyme C:P ratio, while the soil bacterial communities were significantly affected by the soil extracellular enzyme N:P ratio. Compared to the shrubland and forest plantation types, C restriction was more severe in the meadow plantation type, while P restriction was observed in all three plantation types. In this study, we selected the

month of July for sample collection. Different seasons and interannual changes might have different driving mechanisms. In future studies, we will further explore the stoichiometric characteristics of soil and extracellular enzyme activities in different vegetation types under four seasons in order to provide a scientific basis for revealing the stoichiometric regulation mechanism of soil and extracellular enzymes in Luya Mountain.

**Author Contributions:** T.J. conceived and designed the experiments. X.W., M.Z. and Y.Z. performed the experiments. B.C. contributed new reagents. X.W., T.J., L.Z. and Y.C. wrote the manuscript. All authors have read and agreed to the published version of the manuscript.

**Funding:** This study was supported by the National Natural Science Foundation of China (grant no. 32171524), Shanxi Province Science and Technology Innovation base construction project (grant no. YDZJSX2022B001), Scientific and Technological Innovation Programs of Higher Education Institutions in Shanxi (grant no. 2019L0005), and Shanxi Province Foundation for Returnees (grant no. 2021-018).

**Conflicts of Interest:** The authors declare that they have no competing financial interests or personal relationships that could have appeared to influence the work reported in this paper.

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
