# Peer review of "Stoichiometric Soil Microbial and Enzymatic Characteristics under Three Different Plantation Types in China’s Luya Mountain"

_forests, doi:10.3390/f14030558_

Round 1

Reviewer 1 Report

Add the reference  and details for S type

Add references in materials and method section especially 2.2 to 2.6

In the section 3.6.  author should elaborate the   metagenomics study, mentioned the total OTUs number and what was the cutoff ,through which you have make the  further analysis.

Add a table and display the statistical  value of  alpha and  beta diversity

Author should also add a comparative  diagram  or statics of  microbial communities (bacteria and fungi)  in-between all the three sampling sites  C. lanceolate meadow, the C. jubata shrub  and the P. asperata forest

Figure quality of all are very poor, no any text are visible and must be improved

Author Response

1.Add the reference and details for S type

Response: We sincerely thank the reviewer for careful reading. As suggested by the reviewer, references have been added and highlight in yellow directly on the revised copy of our manuscript. We learned from the following article. Jia Tong, Guo Tingyan, Yao Yushan, Wang Ruihong, Chai Baofeng. Seasonal microbial community characteristic and its driving factors in a copper tailings dam in the Chinese Loess Plateau. Frontiers in Microbiology. 2020. 11: 1574.

2.Add references in materials and method section especially 2.2 to 2.6

Response: Thank you for the great suggestions. We have added references to the manuscript and have highlighted them in yellow. Specific references are listed below.

  1. Jia, T.; Guo, T. Y.; Yao, Y. S.; Wang, R. H.; Chai, B. F. Seasonal Microbial Community Characteristic and Its Driving Factors in a Copper Tailings Dam in the Chinese Loess Plateau. Front microbiol, 2020, 11. https://doi.org/10.3389/fmicb.2020.01574
  2. Spohn, M.; Klaus, K.; Wanek, W.; Richter, A. Microbial carbon use efficiency and biomass turnover times depending on soil depth - Implications for carbon cycling. Soil biol biochem, 2016, 96, 74-81. https://doi.org/10.1016/j.soilbio.2016.01.016
  3. Carreiro, M. M.; Sinsabaugh, R. L.; Repert, D. A.; Parkhurst, D. F. Microbial enzyme shifts explain litter decay responses to simulated nitrogen deposition. Ecology, 2000, 81(9), 2359-2365. https://doi.org/ 10.1890/0012-9658(2000)081[2359:MESELD]2.0.CO;2
  4. Jia, T.; Wang, Y. W.; Chai, B. Bacterial Community Characteristics and Enzyme Activities in Bothriochloa ischaemum Litter Over Progressive Phytoremediation Years in a Copper Tailings Dam. Front microbio, 2020, 11. https://doi.org/10.3389/fmicb.2020.565806
  5. Jia, T.; Wang, R.; Chai, B. Effects of heavy metal pollution on soil physicochemical properties and microbial diversity over different reclamation years in a copper tailings dam. J soil water conserv, 2019, 74(5), 439-448. https://doi.org/10.2489/jswc.74.5.439

3. In the section 3.6. author should elaborate the metagenomics study, mentioned the total OTUs number and what was the cutoff, through which you have make the further analysis.

Response: We sincerely thank the reviewer for careful reading. We have supplemented the content you suggested in the manuscript. The details are as follows. After quality screening and minimum sample sequence number extraction, the resulting bacterial or fungal effective sequences were used for analysis, which were clustered into 4847 bacterial OTUs and 2630 fungal OTUs according to 97% similarity.

4. Add a table and display the statistical value of alpha and beta diversity

Response: We sincerely thank the reviewer for careful reading. The statistical value of alpha and beta diversity is already shown in Table 4 and Figure 5.

5. Author should also add a comparative diagram or statics of microbial communities (bacteria and fungi) in-between all the three sampling sites C. lanceolate meadow, the C. jubata shrub and the P. asperata forest

Response: Thank you for the reminding. Comparative analysis of soil microbial communities among different vegetation types is shown in Figure 6 and Figure 7.

6. Figure quality of all are very poor, no any text are visible and must be improved

Response: We sincerely thank the reviewer for careful reading and sorry about the poor picture quality. The picture quality has been changed.

Reviewer 2 Report

The manuscript entitled "Stoichiometric soil microbial and enzymatic characteristics under three different plantation types in China’s Luya Mountain " is of some interest. This work is aimed to provide a theoretical basis for an in-depth understanding of the regulatory role that microbes play within different ecosystems as well as the mechanisms associated with soil enzyme response. There are many factors such as soil nutrients, soil extracellular enzyme, and soil microbial stoichiometry characteristics and their key influencing factors in a Picea asperata forest, a Caragana jubata shrubland, and a Carex lanceolate meadow were analyzed. However, several issues must be addressed as follows:

1) Authors mentioned that the studied area has four main vegetation types but, in the paper, the authors sampled and tested only three types. I am wondering based on which decision only three types were selected?

2) Figure 2 is a bit confusing in the following way – is it correct to do a linear relationship when the dots have a cluster-distribution-manner? Additionally, it is much evidence that there is hard to draw a line between all these dots.

3) I would recommend modifying the introduction section. I am still wondering why there is so important to investigate the soil systems located in The Luya Mountain National Nature Reserve in Ningwu County. Which significant and unique features do these soil systems have in comparison with others? 

4) The discussion section also has to be rewritten. Authors should pay much more attention to how exactly the vegetation type of the collected soil systems impacts received differences in soil enzyme activities, bacterial and fungal communities, and other parameters on tested soil samples rather than just counting that they have the same results as other authors.

5) Modify the conclusion by adding quantitative results to it.

Author Response

1. Authors mentioned that the studied area has four main vegetation types but, in the paper, the authors sampled and tested only three types. I am wondering based on which decision only three types were selected?

Response: We sincerely thank the reviewer for careful reading. In this study, the stoichiometric characteristics of forest, shrub and grassland of different vegetation types and the key driving factors affecting these characteristics were investigated in Luya Mountain. The dominant species of forest were mainly Picea asperata and Larix gmelinii, and Picea asperata was chosen as the representative sample of forest in this paper. So we selected these three vegetation types in our study.

2. Figure 2 is a bit confusing in the following way – is it correct to do a linear relationship when the dots have a cluster-distribution-manner? Additionally, it is much evidence that there is hard to draw a line between all these dots.

Response: We really appreciate your careful reading of our articles. Here we learned from a related article in Soil Biology and Biochemistry journal and performed a similar process to get Figure 2, as detailed in the article below.

Yang, Y.; Chao, L.; Yunqiang, W.; Huan, C.; Shaoshan, A.; Scott, X. C. Soil extracellular enzyme stoichiometry reflects the shift from P- to N-limitation of microorganisms with grassland restoration. Soil Biol Biochem, 2020, 149(prepublish). https://doi.org/ 10.1016/j.soilbio.2020.107928

3. I would recommend modifying the introduction section. I am still wondering why there is so important to investigate the soil systems located in The Luya Mountain National Nature Reserve in Ningwu County. Which significant and unique features do these soil systems have in comparison with others? 

Response: Thank you for your careful reading of our articles. We have revised it in the article, and the details are as follows. The Luya Mountain National Nature Reserve in Ningwu County, Xinzhou City, Shanxi Province, China, is rich in plant resources, subsequently playing a key role in water conservation, C fixation, oxygen (O) release, nutrient accumulation (i.e., TN and TP), and soil conservation and biodiversity protection measures. These factors are highly significant for maintaining the functional stability and ecological balance of ecosystems.

4. The discussion section also has to be rewritten. Authors should pay much more attention to how exactly the vegetation type of the collected soil systems impacts received differences in soil enzyme activities, bacterial and fungal communities, and other parameters on tested soil samples rather than just counting that they have the same results as other authors.

Response: We really appreciate your careful reading of our articles. We have added differences in how exactly vegetation type affects soil enzyme activity, bacterial and fungal communities, and other parameters on the soil samples tested.

5. Modify the conclusion by adding quantitative results to it.

Response: We sincerely thank for your suggestion, and we have added quantitative results to modify the article conclusion. Details are as follows.

P. asperata forest soil moisture content was higher by a factor of 2.9% compared to that of the meadow plantation. NH4+-N content in the meadow and forest plantations was 27.01% and 32.29% higher compared to the shrubland plantation, respectively. Shrubland N-acetyl-β-D-glucosidase enzyme activity was highest, namely, greater by factors of 1.07 and 1.06 compared to the meadow and forest plantations, respectively. PPO activity significantly differed among the three plantations types, where it was highest in the shrubland plantation and lowest in the forest plantation.

Round 2

Reviewer 1 Report

Authors have resolved the quarries, article can now be accepted in the present form